# Perspectives of compounding pharmacists on alcohol-based hand sanitizer production and utilization for COVID-19 prevention in Addis Ababa, Ethiopia: A descriptive phenomenology study

**Assefa Mulu Baye**[1]*, **Andualem Ababu**[2], **Regasa Bayisa**[2], **Mahdi Abdella**[2], **Edessa Diriba**[2], **Minyechel Wale**[3], **Muluken Nigatu Selam**[4]

1 Department of Pharmacology and Clinical Pharmacy, School of Pharmacy, Addis Ababa University, Addis Ababa, Ethiopia, 2 Pharmaceutical and Medical Equipment Directorate (PMED), Ministry of Health, Addis Ababa, Ethiopia, 3 All African Leprosy, Tuberculosis Rehabilitation and Training Center (ALERT), Addis Ababa, Ethiopia, 4 Department of Pharmaceutics and Social Pharmacy, School of Pharmacy, Addis Ababa University, Addis Ababa, Ethiopia

* assefa.mulu@aau.edu.et

## Abstract

### Background

Globally, the safety of patients and healthcare providers is at risk due to health care-associated infections (HCAIs). World Health Organization and the Centers for Disease Control and Prevention recommend using alcohol-based hand rub (ABHR) for hand hygiene in healthcare settings to prevent HCAIs. Irrational use of ABHR will have undesirable consequences including wastage of products, exposure of healthcare providers to infections and emergence of microbial resistance to the alcohol in hand sanitizers. This study aimed to explore the perspective and experiences of compounding pharmacists on production and utilization of ABHR solution for coronavirus disease in 2019 (COVID-19) prevention in public hospitals of Addis Ababa, Ethiopia.

### Methods

A descriptive qualitative study using in-depth interview of 13 key-informants serving as compounding pharmacists in public hospitals of Addis Ababa, Ethiopia, was conducted. The study participants were identified and selected by purposive sampling. All transcribed interviews were subjected to thematic analysis and transcripts were analyzed manually.

### Findings

The compounding pharmacists in this study had a mean age of 30.6 (±3.1) years and nine of the thirteen participants were men. Ten participants believed that the compounding practice in their respective sites followed the principles of good compounding practice. More than half of the participants did not believe that ABHR products were used rationally in health facilities. They argued that users did not have enough awareness when and how to use

**Data Availability Statement:** All relevant data are within the manuscript and its Supporting Information files.

**Funding:** The authors received no specific funding for this work.

**Competing interests:** The authors have declared that no competing interests exist.

sanitizers. Most of the interviewees reported that compounding personnel had no formal training on ABHR solution production. Study participants suggested incentive mechanisms and reimbursements for experts involved in the compounding of ABHR solutions.

## Conclusion

Three of the compounding pharmacists indicated that ABHR production in their setting lack compliance to good compounding practice due to inadequate compounding room, quality control tests, manpower and equipment. Despite this, most study participants preferred the in-house ABHR products than the commercially available ones. Thus, training, regular monitoring and follow-up of the hospital compounding services can further build staff confidence.

## Introduction

Globally, safety of patients and healthcare providers is concerning due to health care-associated infections (HCAIs). The consequences of HCAIs include prolonged hospital stay, long-term disability, resistance of microorganisms to antimicrobials, additional financial burdens, massive death, and increased costs for the healthcare system. In developing countries, the magnitude of the problem of HCAIs is estimated to be over 25%, resulting in morbidity and mortality [1, 2].

Hand hygiene is the simplest and least expensive measure proven to be effective in preventing HCAIs [2, 3]. Following the emergence of coronavirus disease of 2019 (COVID-19) pandemic, hand hygiene is getting much attention. Choosing an appropriate method of hand hygiene technique depends on various factors. In situations where availability of hand washing materials and water is limited, alcohol-based hand rub (ABHR) is the best alternative [1].

World Health Organization (WHO) and the Centers for Disease Control and Prevention (CDC) recommend using ABHR for hand hygiene in healthcare settings, unless physical removal of microbes with soap and water is required [3, 4]. As ABHR causes less irritation to hands, is more effective in killing most pathogens, takes less time to use, dries automatically, and can be used at the patient bedside; guidelines favor ABHR over soap and water in most cases [5]. Evidences also indicate that ABHR is associated with a higher hand-hygiene compliance rate [6].

However, problems in supply and high cost of commercially-produced sanitizers highly limit the utilization of ABHRs in developing countries [7, 8]. Cost analysis from a case study conducted in Rwanda revealed a 71% financial savings when producing in-house ABHRs rather than a commercially-bought sanitizer [1]. Moreover, a rapid evaluation of 20 different commercial ABHR formulations by the Ethiopian Standard Agency (ESA) revealed that 70% of the products analyzed contained alcohol below the recommended limit (less than 75%) and all products failed to meet the limit test for hydrogen peroxide content [9].

Currently, due to the COVID-19 pandemic, drug regulatory authorities in different countries are empowering health facilities for in-house compounding of ABHR solutions [10]. The Ethiopian Food and Drug Authority (EFDA) gave temporary production license for more than 100 institutions in the country. Producers are expected to ensure that quality is not compromised and safety of patients and healthcare providers is not negatively affected by the use of substandard materials [11].

Irrational use of sanitizers will have undesirable consequences, particularly during the COVID-19 pandemic, which include wastage of products, exposure of health providers to

infections and emergence of microbial resistance to the alcohol in hand sanitizers [12, 13]. Perspectives of pharmacy professionals on the rational use of ABHR can minimize these undesirable consequences.

Pharmacists are one of the front-line healthcare providers during COVID-19 pandemic; they provide public health services through regular supply of medicines, provision of health information on hand washing techniques, sanitizer preparation and utilization, availability of face masks and instructions for their proper use and disposal [14, 15]. As a result, countries recognized the role of pharmacists in combating the pandemic of COVID-19. In New Zealand, the pharmacist's contribution during this pandemic is appreciated by the government by extra remuneration for their support [16].

Evidences are hardly available that explored perspectives of pharmacists on production and utilization of ABHR; their recognition and reimbursement for their direct involvement in combating COVID-19 and implications on their future professional role in containing the pandemic. Therefore, this study aimed to explore the experience and perception of compounding pharmacists on production and appropriate utilization of ABHR solution for COVID-19 prevention in public hospitals of Addis Ababa, Ethiopia.

## Methods

### Study design

A descriptive phenomenology study based on semi-structured guided interviews with compounding pharmacists was conducted to understand their experience and perceptions on production and utilization of hospital-based ABHR solution for COVID-19 prevention in Addis Ababa. The interview was undertaken between May 28 and June 07, 2020.

### Settings, sampling and participants

A key-informant interview was conducted with pharmacists working as compounding case team coordinators in thirteen public hospitals in Addis Ababa, Ethiopia. Purposive sampling was used to identify and select participants for the study. Addis Ababa is the capital of Ethiopia, with a projected population of approximately 3.6 million in 2019 [17]. All interviews were conducted privately in the compounding pharmacists' office. Compounding pharmacists who had agreed to participate in the study were approached by the research team. The key-informant interview was continued until the thirteenth interview as the information gathered was saturated at this point [18].

### Interview tool

A semi-structured interview guide was used to collect data from key-informants (S1 File). The interview guide was developed based on data from WHO, Ministry of Health-Ethiopia (MoH) and EFDA compounding guidelines and manuals [19–21]. It consisted of open-ended questions, such as "what problems do you face in the supply of ingredients, personnel protective equipment, packaging and labeling materials?", so that it can provide interviewees with maximum opportunity to express their views. The questions in the interview guide were evaluated by the research team in terms of relevance and appropriateness. The interview guides prepared in English were translated into Amharic Language for better understanding by the key-informants. Amharic and English Language experts were asked to verify the translation from English to Amharic Language for its accuracy and for appropriateness of the words used in the translated version.

## Data collection

One-to-one face-to-face in-depth interviews of key-informants were conducted to collect the data. Before the actual interviews, objectives and process of the study were explained to the interviewees by the research team. The key-informants read the participant information sheets and were encouraged to raise questions about the study, which were answered accordingly. Participants gave verbal consent to take part in the study. Audio-recording was done for all the interviews and the researchers managed all interviews while the research assistant took the back-up notes. Each interview lasted for approximately 30 to 45 minutes. The interviews were done in Amharic language and to enrich the study objectives, probing questions were asked where appropriate to capture more detailed information on the issue involved. Demographic data of key informants was obtained before the interviews were conducted. When no additional information was obtained from participants, the content of the interviews was deemed to be saturated. The interviews were transcribed verbatim.

## Data analysis

Transcription of the audio-recorded interview in Amharic language was done. Then it was translated into English by the researchers. All transcribed interviews underwent manual thematic analysis to detect the possible themes [22]. In all these steps, results were thoroughly discussed by research team.

## Study rigor

Researchers considered the credibility, dependability, and transferability of each participant's interview to maintain the rigor of this study. To ensure the credibility, audit trail was conducted throughout the interview to make sure that the interpretation of the researchers was according to the participant's interviews. Identification and selection of the appropriate codes and the respective themes was also ensured by consensus among members of the research team. To ensure dependability of the findings, all interviewees were approached using the same interview guide. Readers can judge the transferability of the study with the proper selection of participants, data collection, and process analysis. We tried to manage reflexivity and minimize our own opinion from influencing the study data by appropriate review of interview transcripts, contrasting codes with the raw data, and comparing the findings with the participants' views repeatedly. Pilot test was conducted at Adama Hospital Medical College to standardize the in-depth interview tool. In reporting study findings relevant elements for reporting qualitative research, COnsolidated criteria for REporting Qualitative research (COREQ) (S2 File) were followed.

## Ethical consideration

Ethical approval was obtained from Ethical Review Board of School of Pharmacy, Addis Ababa University (ERB/SOP/229/06/2020). Consent was obtained verbally and it was approved by the Ethical Review Board. All participants gave consent for excerpts of their interview transcript to be published. Anonymity and confidentiality of study participants and their freedom to leave the interview were assured.

# Findings

## Description of study participants

All the interviewed participants (N = 13) were working as pharmaceutical compounding coordinators in their respective health facilities. The participants had a mean age of 30.6 (±3.1)

years and nine of the thirteen participants were men. With respect to educational status, ten participants had first degree and three participants had master degree as the highest qualification. A participant stated 10 years of total working experience since first employed whereas most had a total of six years or less (7 participants). Three participants reported that ABHR solution production was launched at their facilities before the advent of coronavirus pandemic (Table 1).

## Themes

Six key themes elucidated the perspective of study participants towards facility-based production of ABHR solution to prevent COVID-19 in public hospitals found in Addis Ababa. These themes were the perspectives of compounding pharmacists on (1) supply of compounding materials, (2) standards of practice, (3) production capacity and future plan, (4) rational use of ABHR, (5) roles and expectations of pharmacist on ABHR production, and (6) supports to mitigate potential challenges.

**Supply of compounding materials.** All participants explained that their ABHR solution production was based on WHO formulation one (Ethyl alcohol-based formulation). Most of the participants indicated that ingredients (ethanol, hydrogen peroxide and glycerol) required for the production of ABHR solution were easily available from local suppliers while some stated that shortage of budget, accidental leakage of ethanol during transportation and erratic supply of dispensing bottles are main problems.

Local plastic factories, private wholesalers and donations were the sources of dispensing bottles, as indicated by respondents. Labeling materials were mentioned to be easily available from local stationary suppliers.

Few of the facilities reported that they already have adequate equipment for ABHR solution production as these facilities had established compounding services for non-sterile preparations. Many of the facilities purchased the equipment specifically for the production of ABHR.

**Standards of practice.** Participants were asked to reflect their views on the practice of facility-based ABHR production and comparison of ABHR products from their facilities with the commercially available ones.

**Table 1. Sociodemographic characteristics of the participants (N = 13).**

| Characteristics | | | Frequency (%) |
|---|---|---|---|
| Age (years) | ≤30 | | 7(53.8) |
| | >30 | | 6(46.2) |
| | Mean (±SD) years | | 30.6 (±3.1) |
| Gender | Female | | 4(30.8) |
| | Male | | 9(69.2) |
| Highest qualification | First degree | | 10(76.9) |
| | Second degree | | 3(23.1) |
| Current practice setting | General hospital | | 9(69.2) |
| | Specialized hospital | | 4 (30.8) |
| Working experience (years) | Total work experience | ≤ 5 | 6(46.2) |
| | | >5 | 7(53.8) |
| | In the current position | <1 | 8(61.5) |
| | | ≥ 1 | 5(38.5) |
| ABHR production starting time | Before COVID-19 Pandemic | | 3(23.1) |
| | After COVID-19 Pandemic | | 10(76.9) |

*Good Compounding Practice (GCP)*. On adherence to GCP principles, ten participants had positive perceptions. They believed that the compounding practice in their respective sites followed the principles of GCP. Availability of appropriate premise and supplies, conducting quality control activities and preparation of ABHR solution using the WHO suggested ingredients were among reasons raised for GCP compliance as illustrated by the following statements:

*. . .our facility satisfied the basic requirements of GCP during production. Dedicated premise, available trained personnel, maintaining good personal hygiene and use of quality raw materials are important parameters for GCP. We also confirm the quality of compounded products by checking turbidity of solution, comprehensiveness of label information, alcohol concentration and others. . . (participant 12/M/30 years)*

*. . .while we are preparing ABHR based on the WHO guideline, we are following GCP. We are using the required standard ingredients including distilled water which is obtained from calibrated distiller. We are also wearing all appropriate personal protective equipment (PPE) during production. We clean the compounding premise and production equipment before and after ABHR production regularly. We have documented all ABHR products distributed to the users. . . (participant 13/F/32 years)*

Some participants claimed that the compounding practice lack compliance to GCP principles. Lack of suitable compounding room and inputs for quality control (QC) tests were mainly stated for the non-compliance. They further added insufficient manpower and equipment as causes for lack of standard ABHR compounding practice which could be indicated by the following excerpt:

*. . .I think the compounding practice in our site lack some of the basic requirements for GCP. For example, there is no well-established premise and we are not assuring the strength of hydrogen peroxide and glycerol used for the production of ABHR solution. Equipment like alcoholmeter and thermometer are not regularly calibrated as well. . . (participant 9/M/34 years).*

*Preference of in-house ABHR solution compared with commercial sanitizers obtained from the market*. Most participants preferred sanitizers produced at hospitals than the ones from the market. They believed that sanitizers compounded at health facilities are superior in terms of quality than those obtained in the local market. Reasons mentioned for the preference included facility-based products were not profit oriented and strictly followed WHO's recommendation. They also claimed that their products are cheaper than those manufactured by local companies which could be illustrated by the following excerpt:

*. . .I recommend hospital-based ABHR than the commercial one as the health care facilities are not compounding ABHR for profit. Hospital-based ABHR fulfills all the quality requirements (e.g., 80% v/v of ethanol) compared to most commercial products. In addition, 130 ml ABHR solution is available for sale in our community pharmacy at a price of 30 Birr (less than $1 USD) against 100 Birr ($3 USD) for commercial product of same volume. . . (participant 12/M/30 years)*

Most participants believe that quality and cost difference among facility-based and commercial products resulted from lack of stringent control of the country's regulatory body over the commercial products. Hence, most sanitizers are being sold on streets, as they pointed out.

**Production capacity and future plan.** *Production capacity*. Majority of the participants reflected that they had optimum production capacity that fulfilled the demand of their respective facilities. They described the weekly or monthly production was based on the rough estimation of the quantity of ABHR needed which could be illustrated by the following quotation:

*. . .we are satisfying the facility's demand by compounding 2000 bottles sanitizer of 250 ml capacity in every two weeks. Our product is delivered to more than 1000 staff in our facility and other institutions like MoH and Addis Ababa Police Department. . . (participant 3/M/28 years)*

Some compounding pharmacists stated that they have limited production capacity in their facilities which couldn't meet the institutional demand. Less number of compounding personnel and absence of some ingredients were the reasons claimed for inadequate production which could be illustrated by the following saying:

*. . .we couldn't satisfy the demand of the staff. That is why additional sanitizers were requested and collected from donors. Main reasons of failure to compound ABHR at our facility regularly are inconsistent supply of ingredients and lack of adequate man power. . . (participant 6/M/35 years)*

*Future production plan*. Majority of participants mentioned that the production of sanitizer in the healthcare facilities will continue in the future as illustrated by the following participant:

*. . .the production of ABHR in our facility was started before COVID-19 pandemic and `will continue in the future as well since it is prepared for infection prevention program. Training on ABHR preparation was given to some staff considering continuity of production. . . (participant 4/M/30 years)*

**Rational use of ABHR solution.** Participants reflected their perception on the appropriateness of ABHR utilization in their facilities. When and how to use the solution and handling of the dispensing bottles were the perspectives used to evaluate the rational use of ABHR solution by participants.

*Utilization of ABHR solution*. Both positive and negative perceptions were recorded. More than half of the participants believed that ABHR products were used irrationally in facilities as illustrated by the following excerpts:

*. . .there is an overall awareness problem among ABHR users in our facility. They don't really know when and how ABHR is used. They are using ABHR repeatedly without touching patients or patient surroundings. There is also inappropriate volume of ABHR for a single use (more or less than 2 ml) . . . (participant 13/F/32 years)*

*. . .some of the hospital staff don't know when and how they should use ABHR solution. E.g., they are using it to clean their hands after taking meals, for disinfecting masks and bags, using on wet hands immediately after hand-washing. . . (participant 9/M/34 years)*

Some participants believed that healthcare providers in their hospitals were using sanitizers rationally which could be illustrated by the following excerpt:

*. . .there is rational use of sanitizers by health workers in our facility because they have aware-ness on when and how much to use. We have also reached the users through focal person of each unit during product distribution. . . (participant 8/M/27 years)*

*Handling of dispensers*. Few participants commented that the bottles used for packaging of the ABHR solution were handled properly by users. They assumed that the proper use of dis-pensing bottles was because of the information provided as illustrated by the following excerpt:

*. . . since, we have no sufficient packaging bottles for our community, we cannot substitute the damaged bottles. This is communicated to users especially for front-line staffs. Therefore, they are handling packaging bottles properly. . . (participant 10/M/33 years)*

On the other hand, most participants stated the inappropriate use of dispensing bottles such as request for bottle substitutions or additional bottles which could be illustrated by the following notation:

*. . .we gave a 125 ml capacity bottles for all users at the beginning. But most of them came without them for the refill schedule and their reasons were loss of the bottle, broken caps etc. This shows the level of attention given to the bottles. . . (participant 13/F/32 years).*

*Perception of pharmacists towards information provided on rational use of sanitizers*. Major-ity of participants agreed that they had adequate knowledge about sanitizers and can deliver all the required information for any enquiry including its rational use which could be illustrated by the following excerpt:

*. . .. we, pharmacy professionals, have basic knowledge about rational use of medicines includ-ing sanitizers. In our facility, we provide as much information as we can via social media, leaf-lets and posters on notice boards which are prepared by members of the hospital drug information center. The information provided include what ABHR is, its ingredients, how to use it and precautions during its usage etc.. . . (participant 13/F/32 years).*

On the reverse, few did not perceive that pharmacists are not the only information experts regarding sanitizers use which could be illustrated by the following extract:

*. . . I don't think that pharmacists are the only information providers on sanitizers' rational use. This time everybody knows about sanitizers and has sufficient information on its rational use as this product is repeatedly advertised to the public through different media. . . (partici-pant 8/M/27 years)*

**Roles and expectation of pharmacists.** All the key informants expressed that ABHR for-mulations were exclusively produced by pharmacy professionals in their respective hospitals. Few of the compounding personnel had in-service training on ABHR solution production facilitated by MoH. Most of the interviewees reported that compounding personnel had no formal training on ABHR solution production.

Asked about their view on how they are duly recognized and reimbursed for their public health services to combat COVID-19 in Ethiopia, most of them perceived that pharmacist were committed to their professional role but their contribution was not given due attentions

by the government, media and the general public, which could be demonstrated by the following excerpt:

> . . . it is well known that COVID-19 is a serious health problem and the contribution of pharmacy professionals in saving life during this pandemic is significant. The necessary PPE and other resources for COVID-19 are mainly forecasted, purchased and distributed to other healthcare providers and patients by pharmacy professionals. But the attention given to the role of pharmacy professionals in this regard is not adequate (e.g., not considered for additional benefits like others healthcare providers) because they are not considered as front-line workers . . . (participant 12/M/30 years)

Federal MoH, Addis Ababa Health Bureau, Ethiopian Pharmaceutical Association (EPA) and the pharmacy departments in the hospitals were mentioned for their suboptimal advocacy activities in promoting the role of pharmacists in combating COVID-19.

As stated by the key informants, benefits expected by pharmacy professionals for their services related to COVID-19 pandemic include compensations for risk (as alcohol could have acute and chronic toxicities) like other healthcare providers. This was illustrated by the following excerpt:

> . . .pharmacy professionals should present their evidence of contribution in reasonable way and attention should be given. Officials from MoH and responsible organizations should clearly understand, recognize and acknowledge the role of pharmacy professionals during the pandemic. . . (participant 12/M/30 years)

**Supports to mitigate potential challenges.** All the study participants need supports from various stakeholders to strengthen their ABHR solution production capacity. Most interviewees believed that consistent supply of ethanol and materials for quality control of hydrogen peroxide and glycerol should be made available by those concerned bodies and hospital ABHR production should be supported and monitored by MoH, and its parastatals. This is illustrated by the following quotation:

> . . .we need a means of confirming the strength of hydrogen peroxide and glycerol raw materials. Supply of ethanol to healthcare facilities should be easy and continuous (available from nearby suppliers when required). Standard packaging bottles with appropriate sizes should be made available to us and this can be facilitated by MoH or other relevant stakeholders. . . (participant 13/F/32 years)

Most key-informants also stressed on the need of in-service training and continuous follow-up of ABHR production and other compounding services by MoH and hospital administration, as indicated by the following informant:

> . . .I believe MoH should provide in-service training on ABHR production and provide consistent follow-up to maintain the sustainability of ABHR production and compounding services. Hospital administrators should pay attention for the compounding service by allocating premises and recruiting sufficient personnel. . . (participant 4/M/30 years).

As most study participants mentioned, incentive mechanisms and additional benefits need to be set for experts involved in compounding of ABHR solutions. Most of them also demand support for compounding equipment including large size measuring devices.

*. . .we are compounding mainly COVID-19 related products with limited number of staff and small size measuring devices that result in repeated exposure to alcohol. The ABHR production team is not a member of COVID-19 team in the hospital that disregards pharmacists from COVID-19 related incentives. MoH should guide our hospital management to consider pharmacy professionals as part of COVID-19 team and get all related compensations and incentives for ABHR production. . . (participant 3/M/28 years)*

## Discussion

The present study was conducted in public hospitals where ABHR production has been in practice. All the interviewees were serving as coordinator for compounding service of the respective hospitals. All participants reported that they were following formulation one for ABHR solution production in which ethyl alcohol is the active agent and hydrogen peroxide, glycerol and water are the other ingredients available in the formulation [19]. Ethyl alcohol-based formulation was being used in all facilities since ethanol is found to be cheaper than isopropyl alcohol and easily available from local sources. Similar findings were reported in other studies [1, 2]. Sugar factories were mentioned as consistent sources of ethanol for production of ABHR.

Good packaging practice is an important aspect during manufacturing of pharmaceuticals that preserves the stability and quality of products [23]. Despite this, most participants raised the challenges of obtaining the desired quantity and size of dispensing bottles. Unavailability of appropriate individual size bottles with safety caps is also reported in other studies [24, 25]. This problem might have its own impact on irrational use of ABHR solution. Shortage of supply of bottles might be because of an imbalance between demand and supply since most facilities started ABHR production after the occurrence of COVID-19 pandemic. Local plastic factories could have faced shortage of imported raw materials to produce bottles.

GCP is a recommended standard in the compounding of non-sterile preparations that meets the regulatory standards of producing safe, effective and quality products. It covers various components like premise, personnel, ingredients, compounding, quality assurance and control, sanitation and hygiene, storage, packaging and labeling, stability, distribution and documentation [26–28]. Despite difference in perception of respondents about GCP, most of them felt that productions of ABHR solutions in their facilities are in line with the standards of GCP. Perceived reasons for compliance of GCP include premise availability, using WHO recommended ingredients and PPE availability were among the others mentioned by participants. Their knowledge on GCP might be limited due to inadequate experience of the compounding service in facilities, mainly ABHR. Lack of appropriate training on GCP (both theoretical and practical) was also raised by the participants.

The acceptance and use of ABHRs by healthcare workers is a crucial factor in the success of any hand hygiene program. Regarding the selection of sanitizers, most preferred the in-house products than commercially produced/purchased ones and same response was reported elsewhere [29]. Facility-based products were given more trust in terms of quality as they are not produced with the notion of maximizing profit. Poor control of manufacturers by the country's regulatory body and issuing manufacturing license to non-pharmacy personnel might have a negative effect on quality of sanitizers especially with respect to the alcohol content. Relatively low cost (sometimes available for free) of in-house produced sanitizers is also indicated for their preference compared to those purchased from the market. Reports by other scholars also showed that sanitizers on the market are costly. For example, cost analysis from a case study conducted in Rwanda indicated a 71% financial savings while producing in-house ABHRs compared to those obtained from market [1, 7, 25, 30, 31].

Since the active agent of ABHR solution is alcohol, the efficacy of sanitizers is primarily determined by its alcoholic content. Hence, manufacturers should ensure the alcoholic strength of their products. Some of the interviewees raised their concerns on safety and efficacy of the commercial alcohol-based hand sanitizers as they are observed to contain fragrance and colors. It is documented that such excipients in the formulations may cause allergenic reactions and decrease the effectiveness of the preparations which otherwise should be evaluated [19, 29, 32, 33]. The drug regulatory body of the country has the mandate and responsibility of continuously assessing the quality of such products available in market, regardless of the sources to safe guarding the general public.

WHO strongly recommends local production of ABHRs in health facilities so that its quality and availability can be insured and it provides low cost alternative for similar products [19]. Most participants reported that there is adequate supply of ABHR solution currently for the hospital community. This is a good start for the studied hospitals as one of the greatest challenges for African hospitals has been that of securing an affordable, available and accessible supply of ABHR [24]. Consistent availability and supply of such a product is highly desired as it is one of the products included in the WHO's essential medicine list [34]. The health facilities included in the study had a plan of reaching the public by scaling up their production capacity. Considering the current COVID-19 pandemic, and improved habit of the general public in using sanitizers, the demand for it exceeds the supply. Majority of participants explained that the production of sanitizers in their facilities will not be interrupted even after COVID-19 is controlled globally. Sanitizers are one of the important supplies for preventing infections in health facilities and the continuation of their production will be the basis for hospital-based compounding of other sterile and non-sterile preparations.

Despite its production and availability, rational use of sanitizers should be an issue of concern for its effectiveness in preventing transmissions of infections. As a finding of this study, most participants indicated the irrational use of sanitizers in the hospitals. Such inappropriate utilization of sanitizers could lead to wastage of products, exposure of healthcare providers to infections and emergence of microbial resistance to alcohol [12]. It is important to follow the instructions on use of sanitizers by manufacturers that are usually given on the label of ABHR bottles. Inappropriate use of dispensing bottles was also mentioned by most study participants and a similar problem was also reported for reused dispensers as a result of pump or cap damage [30]. Accessibility of products at the point of care might improve the appropriateness of its use and this can be achieved by using wall-mounted dispensers in proximity to the service delivery places [29]. Use of such dispensers can also minimize mis-handling practice of packaging bottles which are mentioned by majority of respondents.

It is essential that the team in charge of implementing the ABHR production and distribution educate their staff about the correct use of the product. This is the responsibility of pharmacists who are knowledgeable on rational use of drugs including sanitizers to provide specific education and ensure its correct use as mentioned by most of the interviewee.

As per the guideline set by EFDA, the compounding unit should be headed by a registered pharmacist who will be in charge of the overall compounding activity. Adequate number of pharmacy professionals should be involved in compounding activities [21]. All the key informants indicated that pharmacy professionals were engaged in the compounding of ABHR solution even though the number of pharmacy professionals assigned for the activity was not adequate. The pharmacy personnel involved in such activity should take an updating training in basic compounding skill, quality control and hygiene procedures [20]. Most of the interviewees reported that compounding personnel had insufficient formal training on ABHR solution production. Unless proper training is provided for compounding pharmacists, effective production, quality control and utilization of ABHR solutions will be compromised.

Most of the respondents perceived that they were committed for their professional role as pharmacist in combating COVID-19 in the country but their contribution was not given due attentions by government, media and the general public. Pharmacists are the most accessible healthcare provider and can act as an adviser in a public health capacity, increasing community awareness by providing appropriate information, advising on precautionary measures and offering counselling with respect to medicines including sanitizers. Moreover, they are primary supplier of necessary products, and can encourage healthcare providers and other individuals who are likely to be exposed to COVID-19 to wear medical masks and other PPEs, giving advice on when to seek treatment from healthcare facilities [35, 36]. To mitigate the current pandemic of COVID-19, innovative legal extensions have conferred the exploitation of the full potential of pharmacists globally, fostering the limited manpower of the overburdened healthcare system [37].

As a result of significant contribution imparted by pharmacy professionals in combating COVID-19 pandemic, some countries have recognized the role of pharmacists. For example, in New Zealand, the pharmacist's contribution during this pandemic is appreciated by the government with extra remuneration for their support [16]. Incentives and compensations for health risk (as alcohol and hydrogen peroxide could have acute and chronic toxicities) like other healthcare were requested by majority of the key informants.

All study participants suggested supports from MoH, Addis Ababa Health Bureau and other stakeholders to strengthen their ABHR solution production capacity. Consistent supply of ethanol, accessing materials for quality control of raw materials and final product, training and continuous follow-up on ABHR production, incentive mechanisms and reimbursements, availability of compounding equipment and revision of organizational structure and functionality of the compounding case team in hospitals were identified to be points of focus of respective stakeholders.

One of the limitations of this study is that findings from the current study only apply to public healthcare facilities in Addis Ababa. Another shortcoming of this study is that most of the study participants were from the pharmacy team; hence reported results are perspectives from these professionals.

## Conclusion

In conclusion, most study participants preferred the in-house products of ABHR than commercially purchased sanitizers. Few of participants perceived that the compounding practice lack compliance to GCP due to lack of convenient compounding room, QC tests, sufficient manpower and equipment. Most of the participants believed that there is inappropriate use of both ABHR products and dispensing bottles in their facilities. Majority of the interviewees reported that compounding personnel had no in-service training on ABHR solution production. So, on job training on compounding of pharmaceuticals should be devised and delivered to pharmacy professionals engaged in compounding service. Regular monitoring and follow-up of the hospital compounding services is also recommended. Finally, supportive mechanisms including relevant incentives and reimbursements are suggested for compounding pharmacists. Moreover, integrating pharmacists and compounding pharmacists in particular, in the pandemic management team should be part of the national policy to mitigate this pandemic.

## Supporting information

**S1 File. Key informants interview guide.**
(DOCX)

**S2 File. COREQ 32 checklist.**
(DOCX)

**S3 File. Excerpts of the transcripts.**
(DOCX)

## Acknowledgments

The research team is very grateful for the study participants and pharmacy department heads who provided their valuable information. We acknowledge Ethiopian Ministry of Health for its technical support.

## Author Contributions

**Conceptualization:** Assefa Mulu Baye, Minyechel Wale, Muluken Nigatu Selam.

**Data curation:** Assefa Mulu Baye, Muluken Nigatu Selam.

**Formal analysis:** Assefa Mulu Baye, Andualem Ababu, Minyechel Wale, Muluken Nigatu Selam.

**Methodology:** Assefa Mulu Baye, Andualem Ababu, Minyechel Wale, Muluken Nigatu Selam.

**Project administration:** Regasa Bayisa, Mahdi Abdella, Edessa Diriba.

**Supervision:** Regasa Bayisa, Mahdi Abdella, Edessa Diriba.

**Writing – original draft:** Assefa Mulu Baye, Minyechel Wale, Muluken Nigatu Selam.

**Writing – review & editing:** Andualem Ababu, Regasa Bayisa, Mahdi Abdella, Edessa Diriba.

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
