## [Decision Letter · Decision Letter 0]

1 Dec 2020

PONE-D-20-24316

Perspectives of compounding pharmacists on alcohol-based hand sanitizer production and utilization for COVID-19 prevention in Addis Ababa, Ethiopia: A descriptive phenomenology study

PLOS ONE

Dear Dr. Baye,

Thank you for submitting your manuscript to PLOS ONE. After careful consideration, we feel that it has merit but does not fully meet PLOS ONE’s publication criteria as it currently stands. Therefore, we invite you to submit a revised version of the manuscript that addresses the points raised during the review process.

Your article has benefited from three excellent reviews. I found each comment by the reviewers worthy of careful consideration. The reviewers point out several areas to improve the introduction and context of the paper. In addition to that context, it is very important at PLOS ONE to provide clear and transparent information about the methodology and so I recommend focusing on the comments by the reviewers regarding the methodology.  

We look forward to receiving your revised manuscript.

Kind regards,

Jim P Stimpson, PhD

Academic Editor

PLOS ONE

Journal Requirements:

2. Please amend your current ethics statement to address the following concerns: Please explain why written consent was not obtained, how you recorded/documented participant consent, and if the ethics committees/IRBs approved this consent procedure.

Reviewers' comments:

Reviewer's Responses to Questions

**Comments to the Author**

1. Is the manuscript technically sound, and do the data support the conclusions?

Reviewer #1: Yes

Reviewer #2: Yes

Reviewer #3: Yes

2. Has the statistical analysis been performed appropriately and rigorously? 

Reviewer #1: Yes

Reviewer #2: Yes

Reviewer #3: N/A

3. Have the authors made all data underlying the findings in their manuscript fully available?

Reviewer #1: Yes

Reviewer #2: Yes

Reviewer #3: Yes

4. Is the manuscript presented in an intelligible fashion and written in standard English?

Reviewer #1: Yes

Reviewer #2: Yes

Reviewer #3: Yes

5. Review Comments to the Author

Reviewer #1: General comments

This is an interesting and informative manuscript looking closely about perspectives of compounding pharmacist on alcohol-based hand sanitizer production and utilization for COVID-19 prevention. The area is worthy of investigation; as relatively little exploration addresses this specific topic. Thus, this would be of interest to readers.

While navigating through the manuscript, there are some punctuation, capitalization and grammar errors that need to be corrected. And I would recommend the manuscript to be proofread again.

Specific comments to each section of the manuscript

Abstract

1. Would authors give explanation about the intent of using two different technical terms “descriptive” phenomenology study in the title of the manuscript and “exploratory” qualitative study in the abstract section?

2. Better to use “in-depth interviews of key-informants for phenomenological study”

3. It is better to use “Findings” instead of “Results” in the qualitative study.

4. It would be good to mention who are the participants or the key informants at least in the beginning of the paragraphs in the results and conclusion section of the abstract. Just like, “The compounding pharmacists in this study had a mean age of 30.6 (±3.1).”

Introduction

1. Every word should be in non-abbreviated form at its first appearance/ mention like COVID-19 and COREQ) (S2_File) etc.

2. Page 4, Line # 63, authors should delete “is” and rewrite it as “As ABHR has low irritation to hands….”

3. Page 4, Line # 81, authors justified the relevance of the study using evidences. One of the evidence is emergence of microorganisms resistance to the alcohol in hand sanitizers. Would authors support this with additional literatures?

Methods

1. Page 5, Line # 103: the type of sampling for the selection of participants in this qualitative study should be explained.

2. Page6, Line # 108 and 109, Setting section: It is appropriate to mention the total number of public hospitals in Addis Ababa in the setting part. “The recruitment of key informants was continued until the saturation point was reached at 13th interview and no new information was obtained from the subsequent interviews [17].” ….. this implies there are other possible interviewees.

3. Page 5, Line 111 : Interview tool …. Pls mention here about the use of conducting pilot test in the standardization of the interview tool.

4. Reflexivity –the positionality and perspectives of the researchers is missed in the manuscript. So it is good to incorporate it as it will affect or shape the whole research findings in quality study.

5. Page 7, Line # 138-142, so long as you use thematic content analysis, it is quite clear to follow and adhere to the steps in the thematic analysis. Do you feel it is important list down the steps here?

Results …. Findings

1. Page 9, Line 167, Table 1: Sociodemographic characteristics of the participants (N = 13). Authors should limit the number of vertical and horizontal lines as minimum as possible.

2. It is more advisable for qualitative researcher to have more narration of themes followed by peculiar quotes/ excerpts that underpin it.

3. On the other hand, the phrase “…illustrated by the following excerpts.” is recurrently appears in this section. So, better to reword it to enhance the palatability of the manuscript to the reader.

Discussion and discussion

It is well narrated section. However, I would recommend couple of points for authors.

1. The authors should argue strongly through comparing their findings with findings and experiences of other studies in other settings. So, the discussion should be anchored deeply in literatures.

2. The authors should distill recommendations for policy input and present them in the discussion well.

Reviewer #2: Dear Authors,

Thank you very much for allowing me to review this manuscript. Compounding of sanitizers was an essential extended role of pharmacist during COVID 19 pandemic around the globe.

Your article is very interesting and reflects to the necessity of implementing a good compounding practice and adequate pharmacist training. To make a bigger impression on the stakeholders in your country, I would refer and point out how practice has expanded in developed countries to show the example:

https://www.sciencedirect.com/science/article/pii/S1551741120306628?via%3Dihub .

Please refer also to FIP guidelines: https://www.fip.org/files/content/priority-areas/coronavirus/COVID-19-Guidelines-for-pharmacists-and-the-pharmacy-workforce.pdf

In many pharmacy practices, compounding of sanitizers was one the first privilege’s for our profession at the beginning of SARS-COV-2 pandemic. I would refer to that in the article in the introduction.

The compounding standard was implemented in many EU states as a chapter in Pharmacopeia and urgently submitted by drug registrations office due to sanitizers deficit.

Since the role of compounding pharmacists is now more and more limited, pandemic is a very good chance to promote pharmacist professional skills.

The study was performed in 13 hospitals in the capital city. A very small sample of staff members took part in the project to make a larger impact.

Additionally, you mentioned about the role of compounding pharmacists but it seems they had different education levels.

Please explain, why there are different levels of education and mention this in the text please and if education may inflence anwsers. Older pharmacist are much better train in compounding than young ones.

I would like a clear information on that point please.

Do you have an up to date numbers from Ethiopian Statistical Office? The 2007 population and housing census of Ethiopia is very old, and I am sure that many things has changed. What is an average number of produced sanitizers in all this 13 hospitals? Please calculate the volume.

My final question, why did you decided to use this methodology, if this kind methodology is commonly used in nursing and midwifery research mainly?

I am missing the study limitation. I would not make my decision based on such a small number of participants, this was rather to show the problem, not to make an embracing change.

Please kindly update your manuscript references and answer my quires.

Thank you.

Reviewer #3: Concerns:

1. How do you evaluate the content validity of the interview guide? It has to be clearly described

2. While the topic and objective of the manuscript is to explore the perspective and experience of compounding pharmacists on production and utilization of ABHR solution, the interview guide consists of question #15 and 16. For example, Participants were asked to reflect on their believe on how they are duly recognized and reimbursed for their public health services to combat COVID-19 in Ethiopia

3. How many pharmacies were there in Addis Ababa? how many of them were producing ABHR at the time of data collection? how do you select study participants (sampling technique if any) .....need to be clarified so that the readers may evaluate the representativeness of the sample used

4. better to use most recent data on population of AA

5. …."The recruitment of key-informant was continued until the saturation point was reached at 13th interview and no new information was obtained from the subsequent interviews…." VS “ ….When no addition information was found from participants the content of the interviews was deemed to be saturated”......clarify the two different statements. What do we mean saturation point in qualitative study?

6. The statement "On the reverse, few did not perceive that pharmacists are information experts regarding sanitizers use" does not support the excerpt which follows it...needs modification

7. The language used while translating the Amharic tape record or note to English (excerpts) needs revision for grammatic errors

6. PLOS authors have the option to publish the peer review history of their article (what does this mean?). If published, this will include your full peer review and any attached files.

Reviewer #1: No

Reviewer #2: No

Reviewer #3: No

---

## [Author Response · Author response to Decision Letter 0]

15 Jan 2021

Point-by-point response

Reviewer #1

General comments

1) There are some punctuation, capitalization and grammar errors that need to be corrected. The manuscript to be proofread again.

RESPONSE: The manuscript is proofread and punctuations, capitalization and grammar errors are corrected accordingly

Specific comments

Abstract

1) Would authors give explanation about the intent of using two different technical terms “descriptive” phenomenology study in the title of the manuscript and “exploratory” qualitative study in the abstract section?

RESPONSE: To be consistent and appropriate according to the nature of the study, we managed it to be “descriptive” phenomenology study as indicated in the title. 

2) Better to use “in-depth interviews of key-informants for phenomenological study”

RESPONSE: Corrected accordingly in the abstract part of the manuscript

3) It is better to use “Findings” instead of “Results” in the qualitative study.

RESPONSE: Yes, we replaced “Results” by “Findings”

4) It would be good to mention who are the participants or the key informants at least in the beginning of the paragraphs in the results and conclusion section of the abstract. Just like, “The compounding pharmacists in this study had a mean age of 30.6 (±3.1).”

RESPONSE: Thank you, we put descriptions of the key informants as recommended (in the result and conclusion section of the abstract)

Introduction 

1) Every word should be in non-abbreviated form at its first appearance/ mention like COVID-19 and COREQ) (S2_File) e.t.c

RESPONSE: As recommended from the reviewer, the non-abbreviated form of COVID-19 is provided as “coronavirus disease of 2019”. The long form of COREQ is also indicated in the manuscript as ”COnsolidated criteria for REporting Qualitative research” 

2) Page 4, Line # 63, authors should delete “is” and rewrite it as “As ABHR has low irritation to hands….”

RESPONSE: Thank you, “is” is deleted.

3) Page 4, Line # 81, authors justified the relevance of the study using evidences. One of the evidences is emergence of microorganisms, resistance to the alcohol in hand sanitizers. Would authors support this with additional literatures?

RESPONSE: The reference indicated as reference #12 was changed to address the original article. An additional article on the efficacy of hand sanitizers is included as indicated in reference # 13. Therefore, the former reference numbers after ref #12 are relabeled

Methods

1) Page 5, Line # 103: the type of sampling for the selection of participants in this qualitative study should be explained

RESPONSE: Indicated as “purposive” 

2) Page6, Line # 108 and 109, Setting section: It is appropriate to mention the total number of public hospitals in Addis Ababa in the setting part. “The recruitment of key informants was continued until the saturation point was reached at 13th interview and no new information was obtained from the subsequent interviews [17].” ….. this implies there are other possible interviewees.

RESPONSE: The total number of public hospitals in Addis Ababa is 13. In few of these hospitals the number of compounding pharmacists (pharmacy professionals) is beyond the total number of the hospitals. As we reached at the point of saturation of information, we ended to proceed the in-depth interview. If there were no saturation at that point, we could have additional number of interviews with in these hospitals

3) Page 5, Line 111: Interview tool …. Pls mention here about the use of conducting pilot test in the standardization of the interview tool

RESPONSE: As indicated in the “study rigor” section of the manuscript, a pilot test was conducted at Adama General Hospital. The purpose of the pilot test is stated in the revised version of the manuscript. 

4) Reflexivity –the positionality and perspectives of the researchers is missed in the manuscript. So, it is good to incorporate it as it will affect or shape the whole research findings in quality study

RESPONSE: Reflexivity is incorporated in the revised version of the manuscript as” We tried to maintain reflexivity and avoid our own opinion from affecting the study data by precisely reviewing interview transcripts, comparing codes with the raw data, and checking the findings with the participants’ views several times”.

5) Page 7, Line # 138-142, so long as you use thematic content analysis, it is quite clear to follow and adhere to the steps in the thematic analysis. Do you feel it is important list down the steps here?

RESPONSE: We are convinced to omit the steps that were listed in the manuscript as it is clear to follow and adhere to these steps.

Results (findings in the revised manuscript)

1) Page 9, Line 167, Table 1: Sociodemographic characteristics of the participants (N = 13). Authors should limit the number of vertical and horizontal lines as minimum as possible

RESPONSE: The frequency and percentage of the study participants are made in a single column in the revised manuscript

2) It is more advisable for qualitative researcher to have more narration of themes followed by peculiar quotes/ excerpts that underpin it.

RESPONSE: General introduction regarding the list and description of themes was already included. As much as possible, further narration of themes was included in the revised version of the manuscript

3) On the other hand, the phrase “…illustrated by the following excerpts.” is recurrently appears in this section. So, better to reword it to enhance the palatability of the manuscript to the reader.

RESPONSE: Addressed accordingly in the revised version of the manuscript

Discussion

1) The authors should argue strongly through comparing their findings with findings and experiences of other studies in other settings. So, the discussion should be anchored deeply in literatures

RESPONSE: An additional literature is included. Publications are very limited on ABHR utilization and production.

2) The authors should distill recommendations for policy input and present them in the discussion well.

RESPONSE: Recommendations for policy inputs are incorporated in the revised version of the manuscript 

Reviewer #2

1) Two material sources were recommended to refer 

RESPONSE: We have considered one of the sources recommended for the discussion of our manuscript

2) The study was performed in 13 hospitals in the capital city. A very small sample of staff members took part in the project to make a larger impact

RESPONSE: We appreciate your valuable comments. This is a qualitative research so that the number of participants is determined by the adequacy/ saturation of information gathered. 

3) Additionally, you mentioned about the role of compounding pharmacists but it seems they had different education levels. Please explain, why there are different levels of education and mention this in the text please and if education may influence answers. Older pharmacist are much better train in compounding than young ones

RESPONSE: The educational levels of the compounding pharmacists are either first degree or second degree. Any of them could be assigned by the hospital manager to coordinate the compounding pharmacy service usually after they are trained on the services. We have included the age of the study participants with the Quotes they provided.

4) Do you have an up to date numbers from Ethiopian Statistical Office? The 2007 population and housing census of Ethiopia is very old, and I am sure that many things has changed. What is an average number of produced sanitizers in all this 13 hospitals? Please calculate the volume

RESPONSE: Thank you. We made correction for this. In the revised manuscript, the projected population of Addis Ababa is 3.6 million in 2019. To calculate the production volume of each hospital was no the interest of the research, rather to explore if the hospitals were able to produce ABHR by their full capacity inline with their demand. 

5) My final question, why did you decided to use this methodology, if this kind methodology is commonly used in nursing and midwifery research mainly?

RESPONSE: The research question we raised demands this methodology. As the research questions are perspectives of pharmacists on the standards of ABHR production, barriers and challenges faced, perspectives on the rational use of ABHR, expectations … therefore, we believe that these types of questions are better addressed by qualitative methods than quantitative ones. The methods are so far utilized in pharmaceutical fields too‼

6) I am missing the study limitation. I would not make my decision based on such a small number of participants, this was rather to show the problem, not to make an embracing change

RESPONSE: The limitation of the study is already indicated in the final paragraph of the discussion in the manuscript. As stated above number of participants for qualitative studies is determined by the saturation level of information gathered

Reviewer #3

1) How do you evaluate the content validity of the interview guide? It has to be clearly described

RESPONSE: Pilot testing of the tool we developed was employed as the method of content validation. This is stated in the manuscript (under the “study rigor”)

2) While the topic and objective of the manuscript is to explore the perspective and experience of compounding pharmacists on production and utilization of ABHR solution, the interview guide consists of question #15 and 16. For example, Participants were asked to reflect on their believe on how they are duly recognized and reimbursed for their public health services to combat COVID-19 in Ethiopia

RESPONSE: These questions were included after the pilot test of the interview guide. The perspectives of compounding pharmacists on the compounding of ABHR was highly connected to their recognition and reimbursement for their activities. 

3) How many pharmacies were there in Addis Ababa? how many of them were producing ABHR at the time of data collection? how do you select study participants (sampling technique if any) .....need to be clarified so that the readers may evaluate the representativeness of the sample used 

RESPONSE: As indicated in the method section of the manuscript, the study participants were those compounding pharmacists working at “public hospitals” not “pharmacies”. The selection of the study participants was purposive, those key-informants (compounding pharmacists). This is also indicated in the “setting, sampling and participants” part of the manuscript

4) better to use most recent data on population of AA

RESPONSE: Corrected, updated data is included in the revised version of the manuscript, 2019 projected population of Addis Ababa

5) …."The recruitment of key-informant was continued until the saturation point was reached at 13th interview and no new information was obtained from the subsequent interviews…." VS “ ….When no addition information was found from participants the content of the interviews was deemed to be saturated”......clarify the two different statements. What do we mean saturation point in qualitative study?

RESPONSE: The later was indicated to explain the former sentence. In qualitative research, saturation is reached when no additional information was found from previous interviews. Sentence construction is revised in the manuscript.

6) The statement “On the reverse, few did not perceive that pharmacists are information experts regarding sanitizers use” does not support the excerpt which follows it…needs modification

RESPONSE: Thank you very much, the statement is corrected as “On the reverse, few did not perceive that pharmacists are not the only information experts regarding sanitizers use which could be illustrated by the following excerpt…”

7) The language used while translating the Amharic tape record or note to English (excerpts) needs revision for grammatic errors

RESPONSE: Revision is made in the revised version of the manuscript

---

## [Editor Report · Decision Letter 1]

10 Feb 2021

PONE-D-20-24316R1

Perspectives of compounding pharmacists on alcohol-based hand sanitizer production and utilization for COVID-19 prevention in Addis Ababa, Ethiopia: A descriptive phenomenology study

PLOS ONE

Dear Dr. Baye,

Thank you for submitting your manuscript to PLOS ONE. After careful consideration, we feel that it has merit but does not fully meet PLOS ONE’s publication criteria as it currently stands. Therefore, we invite you to submit a revised version of the manuscript that addresses the points raised during the review process.

This article does not meet publication criteria #5 https://journals.plos.org/plosone/s/criteria-for-publication. PLOS ONE does not copyedit accepted manuscripts, so the language in submitted articles must be clear, correct, and unambiguous. I recommend that authors seek independent editorial help before submitting a revision. These services can be found on the web using search terms like “scientific editing service” or “manuscript editing service.”

We look forward to receiving your revised manuscript.

Kind regards,

Jim P Stimpson, PhD

Academic Editor

PLOS ONE

Additional Editor Comments (if provided):

This manuscript needs another round of revisions to clarify the writing. Be as specific as possible and provide the data.

I will illustrate from the Abstract given how important this part of the manuscript is for indexing and dissemination, but this comment extends beyond the abstract to the entire paper.

Page 2 Line 34: Add more details about the methods. For example, how were the pharmacists selected?

Page 2 Line 36-37: “The compounding pharmacists in this study had a mean age of 30.6 (±3.1) years and men were the dominant one.”

Instead, of “men were the dominant one,” you should provide the descriptive stats for what proportion were male.

Lines 37-38: “Most participants believed that the compounding practice with

their respective sites followed the principles of good compounding practice.” Define what “most participants” means. Provide the data.

Lines 41-43: “Study participants demand incentive mechanisms and reimbursements for experts involved in compounding of ABHR solutions.” Did they “demand” incentives or is there a better word choice like “suggested” or “recommended.”

Lines 45-47: “Few of participants the compounding pharmacists indicated that the compounding practice lack compliance of good compounding practice due to lack of convenient compounding room, quality control tests, sufficient manpower and equipment.” This is an awkward sentence that needs significant revision. What does “few of the participants” mean? The word “lack” is used twice which confuses the meaning in the sentence.

Here are more examples from the Methods section, which is also critical for readers to understand the study. Again, this is not an exhaustive list, but illustrates the need for greater attention to the writing quality.

Page 6 Lines 110-111: “The compounding pharmacists who had agreed to participate in the study were approached.” Approached by whom? In what way were they approached? This sentence either needs to be clarified or deleted.

Lines 111-113: “The recruitment of key- informants was continued until the saturation point was reached at 13th interview and as no new information was obtained from the subsequent interviews.” 13th should be spelled out as thirteenth. The word “the” should proceed 13th. This sentence is unclear as it suggests that 13 interviews were conducted but infers that more were conducted but that the subsequent interviews did not yield new information. Therefore, 13 interviews were used for the analysis but how many interviews were conducted?

---

## [Author Response · Author response to Decision Letter 1]

27 Mar 2021

Comment: This article does not meet publication criteria #5 https://journals.plos.org/plosone/s/criteria-for-publication. PLOS ONE does not copyedit accepted manuscripts, so the language in submitted articles must be clear, correct, and unambiguous. I recommend that authors seek independent editorial help before submitting a revision.

Response: In this revision we made all possible correction to make the manuscript clear, and correct, and unambiguous. A senior professor who is expert in the field of the study was invited and made very important corrections. 

Comment: Page 2 Line 34: Add more details about the methods. For example, how were the pharmacists selected?

Response: Further descriptions are included in this revision in the method section of the abstract. 

Comment: Page 2 Line 36-37: “The compounding pharmacists in this study had a mean age of 30.6 (±3.1) years and men were the dominant one.” Instead, of “men were the dominant one,” you should provide the descriptive stats for what proportion were male

Response: The descriptive stats for the proportion of men is indicated with the actual figure as “…mean age of 30.6 (±3.1) years and nine of the thirteen participants were men.”

Comment: Lines 37-38: “Most participants believed that the compounding practice with their respective sites followed the principles of good compounding practice.” Define what “most participants” means. Provide the data.

Response: It is correcting by specifying the actual number of participants that maintain the principle of good compounding practice. It is rewritten as: “Ten participants believed that the compounding practice in their respective sites followed the principles of good compounding practice.” This is also corrected in the main result section of the manuscript. 

Comment: Lines 41-43: “Study participants demand incentive mechanisms and reimbursements for experts involved in compounding of ABHR solutions.” Did they “demand” incentives or is there a better word choice like “suggested” or “recommended.”

Response: Thank you for the “soft words” suggested. We modified the sentence as “Study participants suggested incentive mechanisms and reimbursements for experts involved in the compounding of ABHR solutions.”

Comment: Lines 45-47: “Few of participants the compounding pharmacists indicated that the compounding practice lack compliance of good compounding practice due to lack of convenient compounding room, quality control tests, sufficient manpower and equipment.” This is an awkward sentence that needs significant revision. What does “few of the participants” mean? The word “lack” is used twice which confuses the meaning in the sentence

Response: The exact number (i.e., three) replaced “few” and the whole sentence is reconstructed as follows: “Three of the compounding pharmacists indicated that ABHR production in their setting lack compliance to good compounding practice due to inadequate compounding room, quality control tests, manpower and equipment.”

Comment: Page 6 Lines 110-111: “The compounding pharmacists who had agreed to participate in the study were approached.” Approached by whom? In what way were they approached? This sentence either needs to be clarified or deleted

Response: They were approached by the research team. Revision is made in this way “Compounding pharmacists who had agreed to participate in the study were approached by the research team.“ Participants were approached after the objectives and the process of the study was explained to them, as stated in the data collection section of the manuscript.

Comment: Lines 111-113: “The recruitment of key- informants was continued until the saturation point was reached at 13th interview and as no new information was obtained from the subsequent interviews.” 13th should be spelled out as thirteenth. The word “the” should proceed 13th. This sentence is unclear as it suggests that 13 interviews were conducted but infers that more were conducted but that the subsequent interviews did not yield new information. Therefore, 13 interviews were used for the analysis but how many interviews were conducted?

Response: The word “the” is included before 13th (spelled out as thirteenth). We did thirteen interviews and we got the information collected at this point to be nearly similar to the preceding data. Therefore, we ceased further data collection. We have included all the thirteen interviews in the analysis. The way we stated this sentence is modified as follows: ”The key-informant interview was continued until the thirteenth interview as the information gathered was saturated at this point.”

Comment: Other corrections… this is not an exhaustive list, but illustrates the need for greater attention to the writing quality.

Response: Corrections are made throughout the manuscript to address the clarity of the manuscript.

---

## [Editor Report · Decision Letter 2]

30 Mar 2021

Perspectives of compounding pharmacists on alcohol-based hand sanitizer production and utilization for COVID-19 prevention in Addis Ababa, Ethiopia: A descriptive phenomenology study

PONE-D-20-24316R2

Dear Dr. Baye,

We’re pleased to inform you that your manuscript has been judged scientifically suitable for publication and will be formally accepted for publication once it meets all outstanding technical requirements.

Kind regards,

Jim P Stimpson, PhD

Academic Editor

PLOS ONE
---

## [Editor Report · Acceptance letter]

17 Apr 2021

PONE-D-20-24316R2 

Perspectives of compounding pharmacists on alcohol-based hand sanitizer production and utilization for COVID-19 prevention in Addis Ababa, Ethiopia: A descriptive phenomenology study 

Dear Dr. Baye:

I'm pleased to inform you that your manuscript has been deemed suitable for publication in PLOS ONE. Congratulations! Your manuscript is now with our production department. 

Kind regards, 

on behalf of

Dr. Jim P Stimpson 

Academic Editor

PLOS ONE